# The Association between Child and Parent Psychiatric Disorders in Families Exposed to Flood and/or Dioxin

**DOI:** 10.3390/bs11040046

**Published:** 2021-04-01

**Authors:** Min Hyung Lee, Betty Pfefferbaum, Robert Portley, Vinay Kotamarti, Fatih Canan, Carol S. North

**Affiliations:** 1Department of Psychiatry, University of Texas Southwestern Medical Center, Dallas, TX 78712, USA; vinay.kotamarti@phhs.org (V.K.); fatih.canan@utsw.edu (F.C.); carol.north@utsw.edu (C.S.N.); 2Department of Psychiatry and Behavioral Sciences, College of Medicine, University of Oklahoma Health Sciences Center, Oklahoma City, OK 73104, USA; betty-pfefferbaum@ouhsc.edu; 3Lifewell Behavioral Wellness, Phoenix, AZ 85012, USA; robert.portley@gmail.com; 4The Altshuler Center for Education & Research, Metrocare Services, Dallas, TX 75247, USA

**Keywords:** disaster, family, structured diagnostic interview, PTSD, child

## Abstract

Associations of disaster mental health sequelae between children and their parents have been demonstrated, but not using full diagnostic assessment. This study examined children and their parents after a series of disasters in 1982 to investigate associations of their psychiatric outcomes. Members of 169 families exposed to floods and/or dioxin or no disaster were assessed in 1986–1987 with structured diagnostic interviews. This vintage dataset collected several decades ago provides new information to this field because of the methodological rigor that is unparalleled in this literature. Disaster-related PTSD and incident postdisaster disorders in children were associated, respectively with disaster-related PTSD and incident postdisaster disorders in the chief caregiver and mother. More flood-only than dioxin-only exposed parents reported great harm by the disaster, but neither children nor parents in these two groups differed in incident psychiatric disorders. Although this study did not determine the direction of causal influences, its findings suggest that clinicians working with disaster-exposed families should work with children and adult members together, as their mental health outcomes may be intertwined.

## 1. Introduction

The mental health effects of major disasters have long been a topic of major clinical and research interest, prompting the accumulation of much literature on this topic, in both adult and child survivors. Children, considered particularly vulnerable to psychosocial consequences, have been the subject of a number of recent reports [1,2,3,4,5,6,7,8,9,10]. Few rigorous studies have examined mental health consequences of disaster on children and parents within family units, and even fewer have assessed fathers as well as mothers [11,12,13,14,15,16].

The disaster mental health literature on adults contains many studies that have used methodologically rigorous assessment tools providing full diagnostic assessment of relevant psychiatric disorders, and even more studies using nondiagnostic measures such as symptom scales and assessing mental health constructs not pertaining to psychiatric disorders. The disaster mental health literature on children contains fewer studies using full diagnostic assessment of psychiatric disorders. Very few studies comparing postdisaster mental health effects in children and their parents have applied full diagnostic methods for the adults [13,14,15], and none to date have used full diagnostic assessment for the children as well as their parents.

In late 1982, a series of devastating disasters converged on the St. Louis, Missouri area, creating a unique opportunity to study the mental health effects of multiple disasters on children and their parents, using newly developed diagnostic interviews created by nearby academicians [17,18]. A waste oil byproduct from chemical manufacturing was sprayed on unpaved roads and horse stables to control dust in a semirural area near St. Louis [19,20]. The oil that was spread in these areas was later discovered to be contaminated with dioxin [19]. Soil testing by the Environmental Protection Agency (EPA) found unsafe levels of dioxin confirmed in 14 sites and suspected in another 41 sites. The entire town of Times Beach was officially declared a dioxin contamination site [20].

A series of severe floods swept through the area at about this same time, resulting in five fatalities, evacuation of at least 25,000 persons from their homes, and an estimated $150 million in property damages [17,18,21]. Most of Times Beach was covered with floodwater. Three weeks later, as flood evacuees tried to return to their homes to clean up and repair the flood damage, the Centers for Disease Control (CDC) issued a health advisory warning residents not to return to Times Beach [17,18,20,21]. Its residents relocated into temporary housing scattered over a wide area, The arrival of spring brought more problems to the already devastated communities: floodwaters again covered these same areas, and additional dioxin sites were discovered [20]. Successful permanent relocation of all residents took four years to complete. As the Times Beach community had voted to take itself out of the federal flood insurance program, residents had no prospect of reimbursement for their losses. Ultimately, the town of Times Beach was dis-incorporated and bulldozed [20].

This study of the psychosocial effects of the 1982 series of St. Louis floods and dioxin disasters used structured diagnostic interviews to assess both children and adults within families, thus being the first study to examine and compare systematically diagnosed disaster-related psychiatric disorders between children and their parents. Importantly, this study examines not only disaster-related PTSD and other disaster-related psychopathology in children but also in relation to both parents, in contrast to prior literature that has largely lacked full diagnostic assessment methods and has predominantly focused only on mothers. As previous studies have demonstrated significant associations of postdisaster mental health effects other than psychiatric disorders between children and their parents, the current study was undertaken to determine whether rigorously diagnosed psychiatric disorders in both children and their parents would also be found to be associated. In disaster psychiatry, just as in general psychiatric care more broadly, psychiatric diagnosis is fundamental to selecting and delivering treatment that is most effective for the presenting problem [22,23,24,25].

## 2. Methods

The research participants were members of representative families from several areas of eastern Missouri with severe flooding and dioxin contamination beginning in late 1982. Families with children between the ages of 3 and 18 years at the time of the disaster were drawn from four sources. The first sample (60 families) was selected from the town of Times Beach that was subjected to both floods and dioxin. The second sample (25 families) was selected from Castlewood, Minker-Stout, and Quail Run/Imperial, communities exposed only to dioxin. The third sample (55 families) was selected from the nearby town of Pacific in Franklin County that was designated by records of the Army Corps of Engineers and the Department of Natural Resources as a severely flooded area but not exposed to dioxin, and this community was socioeconomically similar to Times Beach. The fourth sample (29 families) was a comparison group without exposure to these disasters selected from the small community of Catawissa located in the same general area as the disaster sites and with similar socioeconomic status. Trailer parks and city blocks that were most similar to neighborhoods in the disaster-affected areas were oversampled.

For selection of participants from the identified communities, government maps of all dwelling units in the selected communities were used to enumerate consecutive households for random selection using a computer-generated random number list. Personal property and real estate tax rolls were used to identify individuals living at the residences selected. Letters were sent to all selected households informing them of the study and that they would be contacted by study personnel to discuss their participation. Eligible participants were screened to ensure that they were living in the community at the time of the disasters and that they had children between the ages of 3 and 18 living in the household at the time of the disasters. Participants from the comparison community were additionally screened to ensure that they were not affected by dioxin or flood exposure. Households were offered $25 for their participation and children in the study received a certificate of study participation. A total of 87% of eligible selected households participated in the study, and 68% of all eligible children and 58% of all identified adults in the household were interviewed about their disaster experience and assessed for psychiatric disorders. The study sample included a total of 290 children and 272 adults from 169 families.

Power analysis conducted for determination of adequacy of the sample sizes with both children and adults was conducted. For example, for the child sample, estimating differences between exposure groups for the diagnosis of disaster-related PTSD (the diagnosis of greatest interest) assuming a conservative 10% diagnostic incidence in the disaster-exposed group and 0% in the unexposed group, assuming values of α = 0.05 and β = 0.20 with a stipulated effect size of 0.8, a sample of at least 158 is needed for identifying differences, much smaller than the sample size of 290 in this study. Similarly, for the adult sample, assuming a conservative 10% diagnostic incidence of disaster-related PTSD in the disaster-exposed groups and 0% in the unexposed group assuming values of α = 0.05 and β = 0.20 with a stipulated effect size of 0.8, a sample of at least 166 is needed for identifying differences, much smaller than the sample size of 272 in this study.

Institutional review board approval for the study was obtained at Washington University in advance of the study, and all participants provided written informed consent at the time of study enrollment. This research was carried out following the rules of the Declaration of Helsinki of 1975. Data collection commenced in July 1986 and was completed in December 1987. Pairs of interviewers conducted simultaneous interviews with parents and children in their homes, a process lasting approximately three hours.

Adult family members were interviewed about their own disaster experience and disaster-related emotional and psychiatric sequelae. They completed the Diagnostic Interview Schedule/Disaster Supplement (DIS/DS) [26] which provided assessment of full *DSM-III* diagnostic criteria for psychiatric disorders in time frames before and after the disaster and detailed information about disaster exposures and losses as well as about subjective perceptions of the disaster and changes in demographic and other psychosocial circumstances (e.g., employment, income, health status) from before to after the disasters [17,18]. The DIS was demonstrated to have acceptable reliability and validity [27].

Information about children in the family and their disaster-related emotional and psychiatric sequelae was collected with the Diagnostic Interview for Children and Adolescents (DICA) using versions for both child self-report (DICA-C) and parent report on the child (DICA-P) [28]. The DICA was demonstrated to have acceptable interrater reliability and validity [21,28,29] and very good agreement with clinician diagnoses [21,30]. Minors in these families were interviewed about their own psychiatric status with the DICA-C, and a parent (usually the mother) was interviewed to provide data on minor(s) in the family using the parent version of the interview (DICA-P). The sample also included 68 children who in the four years between the time of the disasters and the data collection for this study had aged beyond 18 years, and these children received the DIS but neither the parent nor child version of the DICA.

Data from the child disaster interviews and diagnostic assessments were merged, matching available diagnosis data from the different diagnostic interviews as available from the available diagnostic modules for each. Diagnostic modules of the parent and child DICA interviews included disaster-related and other PTSD, major depression, generalized anxiety disorder, overanxious disorder, separation anxiety disorder, oppositional defiant disorder, conduct disorder, and attention deficit/hyperactivity disorder. Diagnostic modules of the DIS included disaster-related and other PTSD, major depression, generalized anxiety disorder, panic disorder, alcohol use disorder, and drug use disorder. First onset and recency data were collected for each disorder and keyed to the point in time of the disasters, allowing generation of lifetime predisaster, postdisaster, and incident diagnoses. All possible diagnosis data for children were captured in summary variables reflecting the occurrence of psychiatric disorder, but because some diagnoses were not included in the data from some of the structured diagnostic interviews for children, multivariate models using the summary diagnostic variables were adjusted for which interviews were completed, in the form of independent covariates added to the models. Summary variables were created to reflect the presence of child internalizing disorders, including PTSD, major depression, generalized anxiety disorder, panic disorder, overanxious disorder, and separation anxiety disorder, and also child externalizing disorders, including oppositional defiant disorder and attention deficit/hyperactivity disorder.

For summarization of univariate results, data were sorted by level of the group of interest and reported for variables oriented to those groupings. For example, characteristics of families were summarized for the 169 families in the dataset, child-level variables were summarized from the orientation of the 290 children in the dataset, and adult member variables were summarized from the orientation of the 272 adult members of the families in the dataset, as reported in Table 1 describing the characteristics of the families, children, and adult members, respectively. For comparison of psychiatric disorders with disaster exposures, analyses of child data in Table 2 were conducted on data sorted at the level of the individual child (total *N* = 290 observations), and analyses of adults in the families in Table 3 were conducted on data sorted at the level of the individual adults (total *N* = 272 observations). Multivariate models examining the association of child diagnoses with adult diagnoses and other relevant variables in Table 4 were conducted on data sorted at the level of the individual child (total *N* = 290). In sorting the data for analysis at the level of the child, because some families had more than one child, individual adults in the family were represented more than once, i.e., one time for each child. Due to this multiple representation of adult family members in the child-sorted data, characteristics of the adults in this report are presented from the adult-sorted data.

A summary variable for parent was created to represent combined data from mothers and/or fathers, and another summary variable was created for main caregiver representing the top level of hierarchy of adults in the household, representing mothers, fathers if no mother was represented in the family data, and then grandmother or grandfather if no parents were represented in the family data (but no grandfathers were main caregivers in these families as determined by this algorithm).

## 3. Data Analysis

The data analysis was conducted using SAS version 9.4 (SAS Institute, Inc., Cary, NC, USA). Descriptive results are presented using counts, proportions, means, standard deviations, and median values. Comparisons of two dichotomous variables used two-tailed Pearson’s chi-square tests, substituting Fisher’s exact tests for instances of cell sizes <5, and McNemar’s tests were used for comparison of paired dichotomous variables. The level of statistical significance was set at the level of *p* ≤ 0.05.

Four multivariate models were created to predict child psychiatric disorders included as the dependent variable for each model from adult family member psychiatric disorders (independent variable of main interest), adjusting for variables including number of household adults, number of household children, flood and dioxin exposures, child age, and use of the parent version of the DICA interview entered simultaneously into the models. These models used the generalized linear mixed models procedures in SAS (PROC GLIMMIX) with a SOLUTION option for fixed effects in the model statement, which linked children within families using a family ID specifier in a RANDOM statement. The potential for variance inflation was examined to avoid excessive multicollinearity among the independent variables in the models using the/VIF function of PROC REG, which revealed all results with values <1.8.

## 4. Results

The sample for this study included a total of 169 families, 140 from disaster-exposed communities and 29 from disaster-unexposed communities. Table 1 summarizes characteristics of the sample, specifically describing the families, the children, and the adults in the sample. Members of these families in the sample included 290 children who were ages 3–18 at the time of the disaster and 272 adults (mostly parents and occasional grandparents). These families contained a median of 5 members (range, 2–9), and a median of 3 members (range, 2–6) were interviewed. A median of 2 children per family (range, 1–7) were identified and a median of 2 children per family (range, 1–4) were represented in the data set. The 290 children in the sample represent 68% of the 427 children ages 3–18 living in the household at the time of the disaster as identified by interviews of the adults in the family. A median of 2 adults per family (range, 1–6) were identified and a median of 2 (range, 1–4) were interviewed. The 272 adults in the sample represent 58% of all 467 adults in the household at the time of the disaster as identified by the adult interviews. Mothers from almost all of the families and nearly two-thirds of fathers were interviewed. Of the 140 families from the disaster-exposed communities, 82% (*n* = 115) were from flood-exposed communities, 61% (*n* = 85) were dioxin-exposed, and 43% (*n* = 60) were from communities that were both flood- and dioxin-exposed.

Table 1 also provides details of the characteristics of the children in the sample. Of the children interviewed, approximately half were male, and the majority were Caucasian. The children were a median of 10 years of age at the time of the disaster and 14 years of age at the time of the interview. At the time of the interview, half of the children were currently enrolled in school and had a median educational level of 8th grade (range, kindergarten through 14 years of education). Children in the sample had an average of 1.6 adult members in the family. The majority of the children had a mother interviewed and nearly two-thirds had a father represented. The vast majority of the children (255/290, 88%) participated in their own diagnostic interview (DICA-C, *n* = 187 or DIS, *n* = 68) and 248 children had a parent interview about the child (DICA-P); 100% of the children had a child and/or parent interview providing child diagnosis data. 

Table 1 further provides details of the characteristics of the adults in the sample. The adults in the sample consisted almost entirely of mothers and fathers. Almost all of the parents were Caucasian and they were a median of 36 years of age at the time of the disaster (range, 20–69). About half of the parents in the study had graduated high school; parents had a median of 12 years of education (range, 3–18). Most of the parents were currently married. About four out of five families had at least one employed parent.

Prevalence of postdisaster child disorders was compared between parent and child versions of the DICA, among 213 minor children who had both DICA-P and DICA-C data (not shown in tables). Presence of any postdisaster child internalizing disorder was identified more often by child than parent interview (38%, *n* = 58 vs. 15%, *n* = 23; McNemar S = 23.11, *p* < 0.001). Individual internalizing disorders more often identified by child than parent interview were major depression (10%, *n* = 15 vs. 3%, *n* = 4, McNemar S = 6.37, *p* = 0.012), generalized anxiety disorder (17%, *n* = 18 vs. 4%, *n* = 4, McNemar S = 9.80, *p* = 0.002), overanxious disorder (21%, *n* = 29 vs. 6%, *n* = 8, McNemar S = 13.36, *p* < 0.001), and separation anxiety disorder (24%, *n* = 33 vs. 6%, *n* = 8, McNemar S = 20.16, *p* < 0.001). Presence of any postdisaster child externalizing disorder was identified equally by parent (16%, *n* = 25) and child (12%, *n* = 18) interviews; however, the individual child disorder of attention deficit/hyperactivity was more often identified by parent than child interview (10%, *n* = 15 vs. 3%, *n* = 5, McNemar S = 5.56, *p* = 0.018).

Using combined child diagnosis data from all study sources of diagnostic interviews of child disorders (DICA-P for parent interviews about children, DICA-C for interviews of minor children, and DIS for adult children), 26% (*n* = 76) of children in the study had a lifetime predisaster disorder, 44% (*n* = 129) had a postdisaster disorder, and 25% (*n* = 73) had an incident disorder (not shown in tables). The disorder with the most incident cases was overanxious disorder (14%, *n* = 19). Only 6% (*n* = 16) of the children in the sample had disaster-related PTSD. Table 2 presents and compares diagnostic findings for the children in disaster-unexposed and disaster-exposed groups. These groups did not differ in the proportions meeting criteria for any psychiatric disorder before or after the disaster or for incident psychopathology overall. By definition, none of the disaster-unexposed children developed disaster-related PTSD, but 7% (*n* = 16) of the disaster-exposed children did develop disaster-related PTSD, a proportion significantly different from 0. A higher proportion of the disaster-exposed children than their unexposed counterparts had postdisaster conduct disorder, but this was largely a function of conduct disorder already present before the disaster and apparently was largely unrelated to the disaster because there were no incident cases. No other single disorder was associated with disaster exposure.

Comparison of diagnoses between flood-only and dioxin-only exposed children found a higher prevalence of any postdisaster disorder in the flood-only group (49%, *n* = 47 vs. 30%, *n* = 13; χ^2^ = 4.24, *p* = 0.039), but this difference was not reflected in incident disorders (not shown in tables). Comparison of diagnoses between children exposed to one disaster (flood or dioxin) and children exposed to both disasters found significantly more disaster-related PTSD (13%, *n* = 11 vs. 4%, *n* = 5; χ^2^ = 5.99, *p* = 0.014) and postdisaster generalized anxiety disorder (27%, *n* = 22 vs. 12%, *n* = 14; χ^2^ = 7.80, *p* = 0.005) among children exposed to both disasters.

Overall, 21% (*n* = 33) of mothers in the study had a lifetime predisaster disorder, 35% (*n* = 56) had a postdisaster disorder, and 24% (*n* = 38) had an incident disorder; 39% (*n* = 43) of fathers had a predisaster disorder, 28% (*n* = 30) had a postdisaster disorder, and 13% (*n* = 14) had an incident disorder (not shown in tables). There were few incident disaster-related disorders: 4% (*n* = 6) in mothers and 2% (*n* = 2) in fathers. The disorder with the most incident cases was generalized anxiety disorder (19%, *n* = 30 in mothers and 12%, *n* = 13 in fathers). Of note, 33% (*n* = 35) of fathers had a lifetime predisaster alcohol use disorder and there were no incident cases after the disaster in either mother or fathers. Table 3 lists diagnostic information for parents in disaster-exposed and -unexposed groups. Very few diagnostic differences were found between disaster-exposed and -unexposed groups. Both mothers and fathers in the disaster-exposed groups had a higher lifetime predisaster prevalence of major depression compared to the unexposed groups, but the disaster-exposed and -unexposed groups did not differ in postdisaster prevalence or incidence of this disorder. The presence of any incident disorder in a parent was associated with disaster exposure. 

Comparison of diagnoses between flood-only and dioxin-only exposure groups found no significant differences between them for either adults or children in the study (not shown in tables). Comparison of diagnoses between parents exposed to one disaster (flood or dioxin) and parents exposed to both disasters found no significant differences. Parents were asked whether they perceived their overall health had changed since the occurrence of the disasters approximately four years before. One or both parents in about one-fourth (24%, *n* = 41) of disaster-exposed households reported worsened health since the time of the disasters, a significantly higher proportion (4%, *n* = 4) than in parents of households not exposed to the disasters (χ^2^ = 16.73, *p* < 0.001), but no differences in perceived health changes were found between flood-only and dioxin-only exposed parents. More fathers in flood-only exposed households than in dioxin-only exposed households held jobs at present (63%, *n* = 26 vs. 37%, *n* = 15; χ^2^ = 4.70, *p* = 0.043). However, household income was lower prior to the disasters in the flood-only than in the dioxin-only exposure groups (mean = 24,540, SD = 12,093 vs. mean = 30,558, SD = 14,597; df = 116, t = 2.36, *p* = 0.020). One or both parents in 60% (*n* = 41) of the flood-only exposed households perceived that the disaster had caused them a great deal of harm, compared to 40% of the dioxin-exposed households (χ^2^ = 10.34, *p* = 0.001).

Table 4 presents results from four multiple regression models predicting child diagnoses (dependent variable) from adult diagnoses (independent variable of interest), adjusting for number of household members, disaster exposures, child age, and, as needed to accommodate for missing interview data, a variable representing completed DICA-P interview, as well as statistically adjusting for linkage of children within families. In the first model, which included only disaster-exposed children, development of child disaster-related PTSD was associated with chief caregiver disaster-related PTSD independent of the other variables. In this model, child disaster-related PTSD was also positively associated with the number of household adults. In the second model, which also included disaster-exposed children, child disaster-related PTSD was predicted from the mother’s disaster-related PTSD. In this model, development of child disaster-related PTSD was associated with mother’s disaster-related PTSD. Child disaster-related PTSD was also positively associated with the number of household adults.

In the third model in Table 4, child incident diagnosis was associated with chief caregiver incident diagnosis independent of other variables in the model, and child incident diagnosis was also independently associated with younger child age. In the fourth model, child incident diagnosis was predicted from the mother’s incident diagnosis. In this model, child incident diagnosis was also associated with mother’s incident diagnosis independent of the other variables in the model, and child incident diagnosis was also independently associated with younger child age. Of note, additional regression models were tested substituting fathers specifically as the adult of interest, and child diagnoses were not associated with fathers’ diagnoses in any of these models. Additionally, similar models were tested with the addition of the variable reflecting perception of a great deal of harm in a list of independent covariates, and this variable added no significant relationships with child psychiatric disorders.

## 5. Discussion

This study examined the psychiatric and psychosocial effects of two disasters of different types on children in relation to adult family members using structured diagnostic interviews to assess both children and members of their families. Both disaster-related PTSD and any incident psychiatric disorder in children were significantly associated with these two same diagnosis variables, respectively in their main caregivers, controlling for other relevant variables including type of disaster. In these models, the associations of these disorders in children and their main caregivers, however, held only for mothers specifically and was not the case for fathers.

This study’s finding that disaster mental health outcomes in children is linked to their parents’ mental health responses has also been found in other studies. Other studies have reported that children’s posttraumatic stress symptoms, other psychiatric symptoms, and difficulties functioning were associated with parental posttraumatic stress symptoms [11,12,31,32,33,34,35,36]. Prior studies have also reported that children’s posttraumatic stress symptoms, school behavior and academic problems, and/or diminished school functioning were associated with parental postdisaster psychopathology [13,14,15,21]. The current study is the first disaster study to examine the association of child and parent psychopathology using structured diagnostic interviews in children as well as their parents, and it was also the first to show that child and parent psychosocial outcomes were associated specifically examining child and parent postdisaster psychiatric disorders.

A number of studies have found child outcomes to be associated with their mother’s outcomes; few studies, however, examined fathers as well as mothers [12,13,14,15,16,37]. Studies assessing both mothers and fathers, even those using the same methodology and assessment tools [13,14,15], have yielded inconsistent results. For example, only maternal psychopathology was associated with children’s reactions to the United States Embassy bombing in Nairobi Kenya [15]. Only maternal psychiatric disorder predicted behavior changes in children after the September 11 (9/11) attacks; however, both maternal and paternal symptoms after 9/11 were associated with their children’s posttraumatic stress symptoms, suggesting independent influences of maternal and paternal factors on children’s disaster outcomes [14]. Possible distinctive maternal and paternal contributions to child outcomes are further supported by a study of a composite sample of children and parents following natural and manmade disasters in which maternal postdisaster psychopathology was associated with child postdisaster behavior changes and decline in school grades, and paternal postdisaster psychopathology was associated with child disaster-related posttraumatic stress symptoms [13].

Clearly, the existing literature provides strong evidence that children’s and their parents’ mental health outcomes of disasters are related. Some authors have speculated that this relationship is in part a natural function of the influences that parents have on their children in general, mediated through both genetic and environmental pathways [14,15,38,39]. In the postdisaster setting, parents who are struggling with their own emotional adjustment may negatively affect their children’s adjustment through unhealthy role modeling, impaired parenting behaviors, inadequate support of their children, and inability to provide basic resources [14,15,38,39,40,41,42,43,44]. Mothers may have greater influence than fathers on their children’s disaster mental health because of the greater amount of time that mothers spend with their children, their chief caregiver role, and their more nurturing characteristics [15,45,46,47,48].

It is possible that the relationship between child and parent mental health outcomes of disaster is not just a function of how parents may affect their children, but also how children may affect their parents in the postdisaster setting. Especially for parents who are dealing with their own problems and distress after a disaster, their burdens may be increased by the greater needs of their children. The distress of their children may further add to their parents’ distress directly, through being more difficult to parent, and through the realization of not having protected their children from harm [12,49,50,51,52,53,54].

It has been concluded in the general literature that the well-established associations between child and parent mental health represent causal relationships that are bidirectional, i.e., not only may parents’ emotional and behavioral states influence their children’s adjustment, but that there may be reciprocal effects of children’s emotional and behavioral states on their parents. Additionally, there may be reciprocal interactions between children’s and parents’ effects on one another within these two opposite pathways of influence [55,56]. Unfortunately, most studies examining associations between children’s and parents’ disaster-associated psychosocial outcomes used research designs limited to examining associations but not causal effects.

Few studies have used research designs appropriate to examine causal relationships within the associations between children’s and their parent’s disaster mental health outcomes by collecting prospective longitudinal data on child and parent outcomes. For example, studies have found that parents’ posttraumatic stress symptoms at one year were associated with their children’s psychiatric symptoms approximately three years after the disaster, possibly representing a causal relationship between the earlier parental symptoms and their children’s later symptoms [11,35]. A study of the Great East Japan Earthquake found that parent posttraumatic stress symptoms one year after the disaster were associated with child internalizing behavioral problems at two years, suggesting effects of parents’ reactions on their children [11]. Other studies have examined the association of children’s and parents’ symptoms in the opposite temporal direction. One of these studies found that children’s intrusion symptoms at one year contributed to their parents’ intrusion symptoms at approximately three years, possibly representing a causal relationship between the earlier children’s symptoms and their parents’ later symptoms [35]. Another study that found independent prediction of children’s later posttraumatic stress symptoms by both maternal and paternal posttraumatic stress symptoms after the Wenshuan earthquake further reported that the children’s posttraumatic stress symptoms prospectively predicted later posttraumatic stress symptoms in their mothers (but not fathers), suggesting a causal role of parental posttraumatic stress symptoms in the development of posttraumatic stress symptoms in their children [16]. A cross-sectional study of child-parent pairs following an earthquake in Indonesia found evidence that parental posttraumatic stress symptoms may have influenced their children’s general psychological distress but no evidence that children’s posttraumatic stress influenced their parents’ general distress [12].

The current study showed no differences between exposure to flood or dioxin in the development of disaster-related PTSD and incident disorders in either children or their adult household members. Consistent with these findings, two prior studies have also determined that it was predominantly the magnitude and scope of disasters rather than their typology (e.g., natural disaster versus technological accident) that provided the greatest contribution to the development of disaster-related PTSD among survivors [57,58]. However, more parents in households exposed only to floods than in households exposed only to dioxin reported subjective appraisals of great harm caused to them. Therefore, even though these two different types of disaster did not have different psychiatric effects, they did differ in subjective appraisals. It may be that the greater subjective harm perceived from flood is that flood waters produce immediate and tangible physical effects, whereas dioxin is invisible and harmful effects from it such as cancer and other medical problems may not be apparent for many years, perhaps even never [59]. The discordance between reported subjective harm from a disaster and objectively measured postdisaster psychopathology suggests that subjective disaster responses might be sensitive to the effects of different types of disasters in ways not evident in the development of psychiatric illness.

Consistent with these findings, a meta-analysis found little difference in children’s posttraumatic stress reactions to natural vs. human-made disasters; rather, the operative factors were death toll and other related factors such as proximity, personal loss, perceived threat, and distress [60]. Little research has addressed exposure to both natural and manmade events in children [61], and few studies have used consistent methodology and assessment tools to compare children across disaster types with the scientific rigor of the current study. One study used the same survey to assess children exposed to Hurricane Katrina and children seeking asylum in Germany [62]. In that study, the refugee children, who had more severe trauma exposures, also had worse psychological outcomes. In both groups of children, their psychological outcomes were significantly associated with their trauma exposures, but this relationship was more pronounced in the refugee population.

The most noteworthy methodological strengths of this study were the size of the sample (*N* = 562); the high participation rate (87%) of the families in the study; the use of structured psychiatric interviews for both children and adults in the same families, including both mothers and fathers; and the control for the linkage of multiple children in families in the multivariate models. No previous studies have provided structured diagnostic interviews of both parents and their children in a large sample of disaster-exposed families; most of the research on this topic has used symptom scales or measured nondiagnostic constructs.

Most child disaster studies, especially studies examining the associations between parent and child outcomes, have focused on posttraumatic stress outcomes. The finding of an association of any psychiatric disorder, as well as PTSD, between children and parents highlights the importance of assessing the range of psychiatric disorders that children and parents may experience, rather than focusing exclusively on posttraumatic stress outcomes. Additional strengths of this study were the collection of diagnostic data on children using both child and parent interviews. Using parent as well as child interviews assessing child disorders provided more complete information on the children’s internalizing disorders, as it has been well established generally that parent’s underestimate their children’s internalizing disorders relative to their children’s assessment of their own psychopathology [63,64,65]. To our knowledge, this disaster study is the first to demonstrate the parental under-recognition of their children’s internalizing disorders in a disaster-exposed sample using structured diagnostic interviews, although previously noted in an article using pilot data from this study [21]. Additionally, the collection of diagnostic data on both mothers and fathers in this study allowed comparisons of associations of child psychopathology with maternal and paternal psychopathology separately.

Another important strength of this study is the collection of diagnosis data relative to the timing of the disasters, providing not only lifetime and predisaster diagnoses but also postdisaster prevalence and incidence. The importance of incident diagnosis is that if the disorder had been present before the disaster, it could not be a product of the disaster exposure, and postdisaster prevalence includes a high proportion of cases that pre-existed and are likely unrelated to the disaster. This study is therefore unique in providing a large and representative disaster sample of children and their parents with diagnostic interview data in both children and their parents and other adult family members for comparison of child and parent disaster-related disorders.

A major limitation to this study is the number of years elapsed since the collection of the data, thus raising concern about the relevance of the data to current knowledge. Despite important advances in disaster preparedness and response in the three intervening decades that may have influenced survivors’ disaster reactions and recovery, the current study addresses knowledge gaps, especially in associations between child and parent psychopathology and effects of disaster type, that have not been clarified in research conducted since then. In particular, few studies have used diagnostic interviews to assess both fathers and mothers and especially children and have generated inconsistent findings providing a need for this research with its methodological rigor of assessment.

Enthusiasm about the relevance of the current study’s data might be diminished by the changes in the diagnostic criteria over time; however, the disorders remain essentially the same constructs in human psychopathology. Further, there has little change to the criteria of many major disorders such as depressive disordes, although PTSD criteria have evolved considerably [66]. Many aspects of life have changed in the intervening decades, but the raw experience of exposure to disaster trauma is likely more stable than changed over these decades, and the findings likely have pertinence in the present time.

The nonparticipation of more than one-third of family members in the household may have also contributed to underestimation of postdisaster psychopathology because nonparticipants are commonly found to have more psychopathology than participants in research [67,68]. Additionally, the data for this study were collected four years after the disaster, which may have resulted in underestimations of postdisaster psychopathology through memory fading over time; however, other research from another sample from this same disaster conducted three years before the current study also found little to no PTSD [17,18,21,69]. Potential underestimation of psychopathology in this study might help explain why the identified rates of disaster-related PTSD were not higher than 7% in the children and 4% in the parents. Despite the low statistical power in such low numbers with PTSD, these variables demonstrated measurable associations between the children and their parents in multivariate models.

This study’s findings have potentially useful implications for mental health intervention in families with children after disasters. As parents in this and other research studies have been found to underrecognize internalizing symptoms and disorders in their children, the importance of interviewing children directly about their own symptoms after disasters will undoubtedly also be important for the assessment of children in clinical settings. A major finding from this study of psychiatric disorders in children from households in the path of community disasters, demonstrating that disorders in children were associated with psychiatric disorders in their adult family members, suggests the importance of inclusion of the family unit, especially parents, in psychiatric assessment and care for children in community-wide disasters, as their psychopathology may be intertwined. Even though this and other studies have not been able to determine causal mechanisms in these relationships that might inform specific interventional directions, the importance of the family unit in clinical care is clearly evident, with need for comprehensive evaluation and/or formal intervention including individual and/or family focused interventions when appropriate. Family interventions offer an opportunity to identify serious problems in individual family members who may need individual assessment and treatment. Decisions about the choice of individual or family interventions, or both, will be ideally based on the mental health and needs of individual family members and on family functioning. 

Further research is needed to examine the differential influence of maternal and paternal psychopathology on children’s outcomes and to explore comprehensively the factors that may account for those differences. More studies are needed to explore the determinants of potential bidirectional influence including specific dynamics of the parent-child relationship and interactions, aspects of parenting, and the family environment and family functioning in children’s and parents’ outcomes. Additional research is needed to determine specific advantages of child-focused, parent-focused, and/or family based interventions in the disaster context.

In adults in particular, biological underpinnings of PTSD have been demonstrated, which involve the hypothalamus-pituitary-adrenal (HPA) axis, neural circuitry of the limbic system, especially the amygdala, the prefrontal cortex, and the hippocampus, and its connections to the parahippocampal gyrus, orbitofrontal cortex, sensorimotor cortex, thalamus, and anterior cingulate cortex. The neuropathological findings in PTSD reflect excessive activation of the amygdala coupled with an underactive medial prefrontal cortex coupled with insufficient inhibition of strong responses to threat of danger from the amygdala [70,71,72,73]. Biological correlates of child disaster outcomes have also been considered [74]. Further research is needed into neuropathological mechanisms of stress responses and PTSD in children, and into their relationships with these responses in other family members, especially parents.

## Figures and Tables

**Table 1 behavsci-11-00046-t001:** Sample description and characteristics.

**Family Characteristics**	
Total # families in sample	169
Total # household children (ages 3–18) identified	427
Total household children/family: mean (SD)	2.5 (1.3)
Interviewed children: % (*n*/*N*)	68 (290/427)
Interviewed children/family: mean (SD)	1.7 (0.8)
Total # household adults identified	467
Total # household adults/family (age > 18 years): mean (SD)	2.8 (1.2)
Interviewed adults: % *(n*/*N*)	58 (272/467)
Interviewed adults/family: mean (SD)	1.6 (0.5)
Total # household members identified	894
Total household members/family: mean (SD)	5.3 (1.9)
Interviewed household members: % (*n*/*N*)	63 (562/894)
Interviewed household members/family: mean (SD)	3.3 (1.9)
Families with adult family member interviewed by member type: % (*n*/*N*)	
Mother	93 (156/169)
Father	63 (106/169)
Grandmother	4 (7/169)
Grandfather	2 (3/169)
Family disaster exposures: % (*n*/*N*)	83 (140/169)
Floods	68 (115/169)
Floods only	33 (55/169)
Dioxin	50 (85/169)
Dioxin only	15 (25/169)
Dioxin and floods	36 (60/169)
No disaster	17 (29/169)
**Child Characteristics**	
Total # children in sample	290
Male sex: % (*n*/*N*)	53 (149/280)
Current age: mean (SD) years	14.2 (5.1)
Age at time of disaster: mean (SD) years	10.2 (5.1)
Caucasian race	95 (276/290)
Currently in school	55 (159/290)
Current grade in school (kindergarten = 0)	7.2 (4.2)
Adult family members/family represented in child dataset: mean (SD)	1.6 (0.5)
Adult family members represented in child dataset: % (*n*/*N*)	
Mothers	93 (270/290)
Fathers	63 (183/290)
Grandmothers	5 (14/290)
Grandfathers	1 (4/290)
Parent and/or child interview regarding child	100 (290/290)
Parent interview (DICA-P)	86 (248/290)
Child interview (DICA-C or DIS)	88 (255/290)
Both parent and child interview	73 (213/290)
**Adult Characteristics**	
*All adult family members*	
Total # adults in sample (*N*)	272
Adult family members: % (*n*/*N*)	
Mothers	57 (156/272)
Fathers	39 (106/272)
Grandmothers	3 (7/272)
Grandfathers	1 (3/272)
*Parents*	
Total # parents in sample (*N*)	262
Age: mean (SD) years	37.1 (9.5)
Caucasian race	98 (256/267)
Level of education	
Less than high school: % (*n*/*N*)	32 (83/262)
High school graduate: % (*n*/*N*)	52 (136/262)
Education past high school: % (*n*/*N*)	16 (43/262)
Years of education: mean (SD)	11.2 (2.3)
Marital status: % (*n*/*N*)	
Married	85 (222/262)
Divorced/separated	13 (33/262)
Widowed	2 (4/262)
Single (never married)	1 (3/262)
Employment: % (*n*/*N*)	
Either parent currently employed	83 (132/160)
Either parent currently employed full time	74 (120/162)
Either parent unemployed since disaster	72 (118/162)

#: number of.

**Table 2 behavsci-11-00046-t002:** Child psychiatric disorders by disaster exposure.

Child Psychiatric Disorders	No Disaster *N* = 51% (*n*/*N*)	Disaster *N* = 239% (*n*/*N*)
**PTSD**		
Disaster-related ^a^	0 (0/51)	7 (16/218)
All postdisaster	2 (1/51)	11 (23/216)
**Major Depression**		
Predisaster	10 (5/51)	11 (25/233)
Postdisaster	10 (5/51)	15 (35/234)
Incident	6 (3/51)	8 (18/233)
**Generalized Anxiety Disorder**		
Predisaster	6 (2/35)	9 (16178)
Postdisaster	17 (6/35)	17 (30/178)
Incident	11 (4/35)	11 (19/178)
**Panic Disorder**		
Predisaster	0 (0/51)	0 (0/237)
Postdisaster	0 (0/51)	0 (0/239)
Incident	0 (0/51)	0 (0/237)
**Overanxious Disorder**		
Predisaster	5 (2/44)	4 (8/195)
Postdisaster	16 (8/50)	19 (41/213)
Incident	9 (4/44)	10 (19/195)
**Separation Anxiety Disorder**		
Predisaster	7 (3/44)	11 (21/196)
Postdisaster	24 (12/51)	17 (38/222)
Incident	14 (6/44)	9 (17/196)
**Oppositional Defiant Disorder**		
Predisaster	2 (1/47)	1 (3/204)
Postdisaster	6 (3/51)	10 (22/227)
Incident	2 (1/47)	3 (7/204)
**Conduct Disorder**		
Predisaster	0 (0/47)	7 (16/217)
Postdisaster ^b^	0 (0/47)	8 (18/217)
Incident	0 (0/47)	1 (3/217)
**Attention Deficit/Hyperactivity Disorder**		
Predisaster	0 (0/46)	2 (5/206)
Postdisaster	2 (1/51)	10 (22/227)
Incident	0 (0/46)	0 (0/206)
**Alcohol Use Disorder**		
Predisaster	2 (1/51)	1 (2/237)
Postdisaster	2 (1/51)	5 (11/239)
Incident	0 (0/51)	3 (7/237)
**Drug Use Disorder**		
Predisaster	0 (0/51)	1 (2/237)
Postdisaster	0 (0/51)	1 (3/239)
Incident	0 (0/51)	0 (1/237)
**Any Disorder**		
Predisaster	16 (8/51)	28 (68/239)
Postdisaster	39 (20/51)	46 (109/239)
Incident	22 (11/51)	26 (62/239)

^a^*p* = 0.048; ^b^
*p* = 0.050.

**Table 3 behavsci-11-00046-t003:** Parental psychiatric disorders by disaster exposure.

Parental Psychiatric Disorders	Mother		Father		Any Parent	
	No Disaster % (*n*/*N*)	Disaster % (*n*/*N*)	No Disaster % (*n*/*N*)	Disaster % (*n*/*N*)	No Disaster % (*n*/*N*)	Disaster % (*n*/*N*)
**PTSD**						
Disaster-related	0 (0/30)	5 (6/128)	0 (0/15)	2 (2/94)	0 (0/45)	4 (8/222)
All PTSD	3 (1/30)	10 (13/128)	0 (0/15)	5 (5/94)	2 (1/45)	8 (18/222)
Predisaster	0 (0/30)	4 (5/128)	0 (0/15)	2 (2/94)	0 (0/45)	3 (7/222)
Postdisaster	3 (1/30)	7 (9/128)	0 (0/15)	3 (3/94)	2 (1/45)	5 (12/222)
Incident	3 (1/30)	6 (8/128)	0 (0/15)	3 (3/94)	2 (1/45)	5 (11/222)
**Major Depression**						
Predisaster	24 (7/29)	6 (8/125) ^a^	13 (2/15)	1 (1/93) ^b^	20 (9/44)	4 (9/218) ^c^
Postdisaster	21 (6/29)	15 (19/126)	13 (2/15)	5 (5/93)	18 (8/44)	11 (24/219)
Incident	0 (0/29)	9 (11/125)	0 (0/15)	4 (4/93)	0 (0/44)	7 (15/218)
**Generalized Anxiety Disorder**						
Predisaster	7 (2/30)	3 (4/128)	13 (2/15)	3 (3/94)	9 (4/45)	3 (7/222)
Postdisaster	10 (3/30)	24 (31/128)	7 (1/15)	13 (12/94)	9 (4/45)	19 (43/222)
Incident	10 (3/30)	24 (31/128)	7 (1/15)	13 (12/94)	9 (4/45)	19 (43/222)
**Panic Disorder**						
Predisaster	0 (0/30)	0 (0/126)	0 (0/15)	0 (0/94)	0 (0/45)	0 (0/220)
Postdisaster	0 (0/30)	1 (1/128)	0 (0/15)	0 (0/94)	0 (0/45)	0 (1/222)
Incident	0 (0/30)	1 (1/128)	0 (0/15)	0 (0/94)	0 (0/45)	0 (1/222)
**Alcohol Use**						
Predisaster	0 (0/30)	4 (5/127)	20 (3/15)	35 (32/92)	7 (3/45)	17 (37/219)
Postdisaster	0 (0/30)	4 (5/128)	13 (2/15)	18 (17/94)	4 (2/45)	10 (22/222)
Incident	0 (0/30)	0 (0/127)	0 (0/15)	0 (0/92)	0 (0/45)	0 (0/219)
**Drug Use**						
Predisaster	3 (1/30)	4 (5/126)	0 (0/15)	8 (7/91)	2 (1/45)	6 (12/217)
Postdisaster	0 (0/30)	0 (0/125)	0 (0/15)	1 (1/91)	0 (0/45)	0 (1/216)
Incident	0 (0/30)	0 (0/126)	0 (0/15)	0 (0/91)	0 (0/45)	0 (0/217)
**Any Diagnosis**						
Predisaster	33 (10/30)	18 (23/128)	40 (6/15)	39 (37/94)	36 (16/45)	27 (60/222)
Postdisaster	27 (8/30)	38 (48/128)	27 (4/15)	28 (26/94)	27 (12/45)	33 (74/222)
Incident	10 (3/30)	27 (35/128)	7 (1/15)	14 (13/94)	9 (4/45)	22 (48/222) ^d^

Compared to no disaster: ^a^
*p* = 0.009, ^b^
*p* = 0.050, ^c^
*p* < 0.001, ^d^ χ^2^ = 3.87, *p* = 0.049.

**Table 4 behavsci-11-00046-t004:** Multiple regression models predicting child disorders from adult disorders ^a^.

Model Variables	Number df	Den df	Estimate	SE	Confidence Limits	T	*p*
**Model 1.** Predicting child disaster-related PTSD from chief caregiver disaster-related PTSD ^b^							
Chief caregiver disaster-related PTSD	1	83	0.22	0.08	0.08, 0.37	20.71	**0.008**
# household adults	1	83	0.04	0.02	0.01, 0.06	20.33	**0.022**
# household children	1	83	−0.01	0.01	−0.03, 0.02	−0.57	0.571
Flood exposure	1	83	0.07	0.05	−0.02, 0.11	10.26	0.211
Dioxin exposure	1	83	0.05	0.04	−0.03, 0.10	10.14	0.258
Child age	1	83	0.00	0.00	−0.01, 0.01	−0.01	0.991
DICA-P interview	1	83	−0.04	0.05	−0.12, 0.06	−0.64	0.523
**Model 2.** Predicting child disaster-related PTSD from mother’s disaster-related PTSD ^b^							
Mother’s disaster-related PTSD	1	78	0.22	0.09	0.05, 0.39	20.52	**0.014**
# household adults	1	78	0.04	0.02	0.00, 0.07	20.28	**0.026**
# household children	1	78	−0.01	0.01	−0.04, 0.02	−0.58	0.566
Flood exposure	1	78	0.08	0.06	−0.04, 0.19	10.29	0.200
Dioxin exposure	1	78	0.05	0.05	−0.04, 0.14	10.16	0.251
Child age	1	78	0.00	0.00	−0.01, 0.01	−0.08	0.937
DICA-P interview	1	78	−0.04	0.06	−0.15, 0.08	−0.62	0.536
**Model 3.** Predicting child incident disorder from chief caregiver incident disorder							
Chief caregiver incident disorder	1	119	0.16	0.06	0.03, 0.28	20.52	**0.013**
# household adults	1	119	0.03	0.02	−0.02, 0.07	10.04	0.300
# household children	1	119	0.02	0.02	−0.02, 0.06	0.96	0.338
Flood exposure	1	119	0.06	0.06	−0.06, 0.17	0.98	0.330
Dioxin exposure	1	119	−0.01	0.05	−0.12, 0.10	−0.19	0.846
Child age	1	119	−0.01	0.01	−0.02, 0.00	−20.20	**0.030**
DICA-P interview	1	119	−0.02	0.08	−0.18, 0.13	−0.32	0.747
**Model 4.** Predicting child incident disorder from mother’s incident disorder							
Mother’s incident disorder	1	112	0.17	0.07	0.04, 0.30	20.53	**0.013**
# household adults	1	112	0.01	0.02	−0.04, 0.06	0.37	0.714
# household children	1	112	0.01	0.02	−0.03, 0.06	0.57	0.571
Flood exposure	1	112	0.06	0.06	−0.06, 0.18	0.96	0.340
Dioxin exposure	1	112	−0.01	0.06	−0.12, 0.11	−0.16	0.874
Child age	1	112	−0.01	0.01	−0.02, 0.00	−20.02	**0.046**
DICA-P interview	1	112	−0.02	0.08	−0.18, 0.14	−0.22	0.822

^a^ Adjusting for number of household members, disaster exposures, child age, and, as needed to accommodate for missing interview data, variables representing completed interviews; ^b^ Model includes only disaster-exposed children; SE: standard error. #: number of. **Bold** font signifies statistically significant *p* values.

## Data Availability

The data from this study are not publicly available.

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
