# Peer review of "The Association between Child and Parent Psychiatric Disorders in Families Exposed to Flood and/or Dioxin"

_behavsci, 2021, doi:10.3390/bs11040046_

Round 1

Reviewer 1 Report

This is interesting and unique data.

I think it would be valuable to describe in the abstract that the data are from 1987.

I lack an explanation about why it took 33 years to submit this manuscript.

Strengths
Background: It is well written, but rather as in a newspaper than in a scientific paper. I lack references.
Method: The methods are well described.
Results: The results are easy to follow for the reader.
Discussion: I think it is well written

Weaknesses
Affiliations: I lack information about country etc.
Abstract: This is relevant, but historical, data. I think that this should be written in the abstract.
Background: I lack a reference(s) for lines 50-79.
Method: I lack a power calculation that argue for the number of included families.
The ethical considerations are weakly described, and this need to be more clarified.
Old material can still be scientifically important, but it would be valuable to describe why it took 33 years before this material was submitted to a journal. Some data are not that relevant today. DSM-III is today DSM-V.
Results: In the result, please add numbers to the percentages (i.e., X % (n=X)).

Author Response

*Please see attachment*

Reviewer 2 Report

I think that the introduction should be modified in order to be less "story telling" and more scientific oriented. I appreciate the detail of the story of St. Louis, however, I do not believe a it is a task for a scientific action to use such narrative. I would appreciate a more summarized description of the events and their consequences (4-5 lines, perhaps) and paragraphs giving us a better in-depth into the main variable of study, main discoveries of previous literature, details on previous studies' methodological approach and main conclusions. This way, we readers would be more aware on why studying such a variable with the proposed approach makes sense.

I wonder how this study gathered data in 1986 and publishes it now. To my eyes, this is a huge flaw. The society has changed over the time, people have changed, approach to safety has changed, household behaviors and relations have changed...I find hard to see a concrete impact of these results on the society. Was it not possible to perhaps gather data on the same people currently, and see the changes from the events to present?

Perhaps as a consequence of the dated data, also the reference list shows a predominance of studies published 10 years ago or older. I think more recent research should be used both for introduction and discussion, regardless of when the data was collected.

Reviewer 3 Report

This is an interesting manuscript investigating the association between child and parent psychiatric disorders in families exposed to flood and/or dioxin, compared to families who were not exposed to disasters.

In the introduction section, the authors have reviewed some aspects with regard to the mental health effects of major disasders on adults, and they discuss that some poapers use rigorous assessment tools. However, assessment scales used to measure symptoms pertaining to psychiatric disorders are scarse. I would also add a brief paragraph based on some biological underspinnings of stress. It would help the reader to better understand why is important to study the psychological consequences of disasters and which biological outcomes they have. Despite most of them are focused on terrorism, there are many works on this topic that apparently test biological hypothesis. This should be brief with 2-3 references.

The methods section is well-presented, including all the information about statistical analyses. Which software did the authors use for statistical analyses? This should be included in this section.

In the results section, the authors reported that they included a total of 169 families: 140 disaster-exposed communities and 29 from unexposed communities. Did the authors consider that comparison should be biased by the unbalanced samples? Disaster-groups is really higher compared to unexposed group. At least, it should be mentioned in the limitations section.

At the discussion section, the authors concluded that the existing literature reports evidence for a relationship between parents’ and children’s mental health outcomes. They also discussed several factors influencing the potential “causal directionality” in the relationship between child and parent mental health outcomes. I would prefer to discuss a “lineality” or association.
